# Epidermal retinol dehydrogenases cyclically regulate stem cell markers and clock genes and influence hair composition
Kelli R. Goggans [1], Olga V. Belyaeva[1], Alla V. Klyuyeva[1], Jacob Studdard[1], Aja Slay[1], Regina B. Newman[2], Christine A. VanBuren[2], Helen B. Everts [2] ✉ & Natalia Y. Kedishvili [1] ✉

The hair follicle (HF) is a self-renewing adult miniorgan that undergoes drastic metabolic and morphological changes during precisely timed cyclic organogenesis. The HF cycle is known to be regulated by steroid hormones, growth factors and circadian clock genes. Recent data also suggest a role for a vitamin A derivative, all-*trans*-retinoic acid (ATRA), the activating ligand of transcription factors, retinoic acid receptors, in the regulation of the HF cycle. Here we demonstrate that ATRA signaling cycles during HF regeneration and this pattern is disrupted by genetic deletion of epidermal retinol dehydrogenases 2 (RDHE2, SDR16C5) and RDHE2-similar (RDHE2S, SDR16C6) that catalyze the rate-limiting step in ATRA biosynthesis. Deletion of RDHEs results in accelerated anagen to catagen and telogen to anagen transitions, altered HF composition, reduced levels of HF stem cell markers, and dysregulated circadian clock gene expression, suggesting a broad role of RDHEs in coordinating multiple signaling pathways.

The hair follicle (HF) is a complex skin miniorgan that changes profoundly during cycles of growth (anagen), apoptosis-mediated regression (catagen), and quiescence (telogen)[1]. The HF cycle is an infradian rhythm, a biological rhythm longer than 24-h[2]. Abnormal regulation of HF cycle control genes was implicated in various types of human hair growth disorders and skin cancers[3]. Although several pathways, including signaling through transforming growth factor beta (TGF-β)/bone morphogenetic protein (BMP), fibroblast growth factors, steroid hormone receptors, and circadian clock genes[4,5], have been shown to play a role in the HF cycle, a complete picture of all the regulatory factors and their crosstalk is far from clear.

Adequate vitamin A levels are essential for hair, as insufficient or excessive amounts of dietary retinoids and topical retinoid treatments may cause alopecia (hair loss) [reviewed in ref. 3]. Vitamin A regulates cell proliferation and differentiation through its bioactive form, all-*trans*-retinoic acid (ATRA), which serves as an activating ligand for transcription factors, retinoic acid receptors (RARs) α, β, and γ. The cellular concentration of ATRA is tightly regulated through multiple mechanisms (Fig. 1) [reviewed in ref. 6]. The uptake of serum all-*trans*-retinol bound

to retinol binding protein 4 (RBP4) by epidermal cells occurs through the membrane receptor stimulated by ATRA 6 (STRA6). Inside the cells, retinoid metabolism is controlled by lecithin retinol acyltransferase (LRAT), retinyl ester hydrolase (REH), retinoid binding proteins, retinoid oxidoreductases, and cytochrome p450 enzymes. LRAT esterifies retinol bound to retinol binding protein 1 (RBP1) to retinyl esters for storage. When needed, REH hydrolyzes retinyl esters back to retinol. STRA6, LRAT and RBP1 can be upregulated by ATRA. Retinoid oxidoreductases control the flux of retinol to ATRA by regulating the levels of retinaldehyde, the immediate precursor of ATRA. The reversible oxidation of retinol to retinaldehyde is the first and rate-limiting step of ATRA biosynthesis[7]. In most cells, this step is catalyzed by the microsomal hetero-oligomeric retinoid oxidoreductase complex composed of two members of the short-chain dehydrogenase/reductase superfamily (SDR) of proteins, i.e., RDH10 (SDR16C4) and dehydrogenase reductase 3 (DHRS3 or SDR16C1)[8,9]. When ATRA exceeds the optimal range specific for each cell type and physiological conditions, DHRS3 is upregulated by ATRA to reduce the flux from retinol to retinaldehyde [reviewed in ref. 10].

[1]Department of Biochemistry and Molecular Genetics, Heersink School of Medicine, University of Alabama at Birmingham, Birmingham, AL, USA. [2]Department of Nutrition and Food Sciences, Texas Woman's University, Denton, TX, USA. ✉e-mail: heverts@twu.edu; nkedishvili@uab.edu

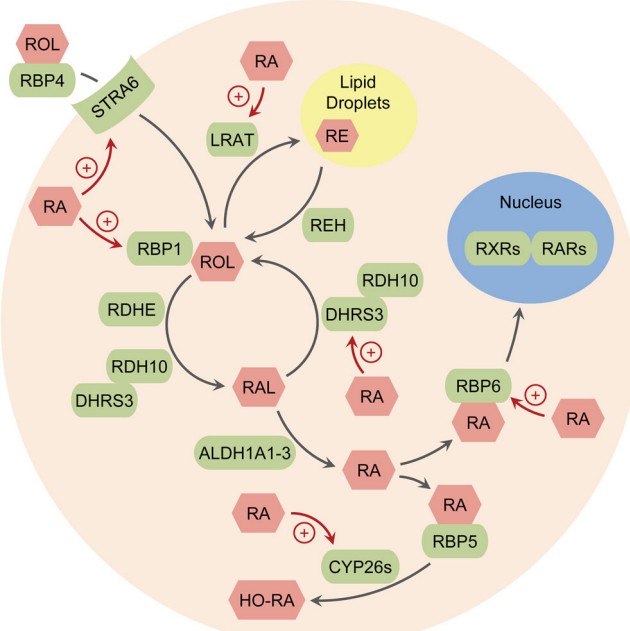

**Fig. 1 | A diagram of retinoid metabolism and signaling.** Proteins are depicted in green and metabolites—in pink. ROL all-*trans*-retinol, RAL all-*trans*-retinaldehyde, RA all-*trans*-retinoic acid, HO-RA hydroxylated RA, RE retinyl esters, STRA6 stimulated by *trans*-retinoic acid 6, LRAT lecithin retinol acyltransferase, REH retinyl ester hydrolase, RBP1 retinoid binding protein 1 (or CRBP1), RDH10 retinol dehydrogenase 10, DHRS3 dehydrogenase reductase 3, RDHE retinol dehydrogenases epidermal, ALDH1A1-3 aldehyde dehydrogenases 1A1, 1A2, and 1A3, CYP26s cytochromes P450 26A1 and 26B1, RBP5 and RBP6 retinoid binding proteins 5 and 6 (or CRABP1 and CRABP2, respectively). Upregulation of specific proteins by RA is indicated by plus signs.

All-*trans*-retinaldehyde is oxidized irreversibly to ATRA by aldehyde dehydrogenases (ALDHs) of the ALDH1A family [reviewed in ref. 11]. ATRA was shown to regulate ALDHs' expression in certain types of cells[6]. ATRA bound to cellular retinoic acid binding protein (CRABP) type 2 (or RBP6) is transported to the nucleus to activate RARs, whereas ATRA bound to CRABP type 1 (or RBP5) is believed to be channeled to CYP26A1 and CYP26B1 for degradation[12,13]. Increased ATRA upregulated the expression of RBP6, CYP26A1 and CYP26B1 via a negative feedback regulation loop[10,14].

Interestingly, previous studies showed expression patterns of retinoid metabolizing enzymes and binding proteins such as ALDH1A1-3, RBP1 and RBP6 undergo dynamic changes in a spatial and temporal pattern during the HF cycle[15,16]. Knowing these genes are regulated by ATRA, the changes in their expression patterns could potentially reflect fluctuations in ATRA signaling. This, however, is currently unknown.

Postnatal cycling and regeneration of HFs are controlled by two major counteracting signals: the BMP and Wnt/β-catenin signaling pathways[17–19]. High BMP signaling keeps HF stem cells (HFSC) in an inactivated state, while Wnt/β-catenin signaling promotes HFSC activation and maintains HF growth. Continued activation of Wnt/β-catenin signaling in dermal papilla through epithelial-mesenchymal interaction is indispensable for anagen progression[20], modulating gene transcription through binding to transcription factors of the TCF/LEF family[21]. In this respect, there is evidence for mutual antagonism between β-catenin and ATRA signaling[22]. β-Catenin upregulates CYP26A1, which causes ATRA degradation and, conversely, β-catenin appears to compete with ATRA-RAR for TCF binding sites[22,23]. Thus, decreased ATRA signaling might be necessary to facilitate the transition from telogen to anagen. The molecular mechanisms decreasing ATRA signaling in skin are unclear. This regulation might include notch signaling, as activation of notch signaling in mouse HFs induced RA metabolism and target genes[24].

HF cycle is also controlled by molecular clock genes[5]. Circadian 24-h rhythms are regulated by positive and negative regulatory feedback loops [reviewed in ref. 25]. The positive feedback loop is controlled by transcription factors CLOCK and BMAL1, which form a heterodimer. Two of their targets, PER (PER 1, 2 and 3) and CRY (CRY1 and 2), form heterodimers and repress their own transcription via direct interactions with CLOCK/BMAL1. Other targets are RORs (*Rora*, *Rorb*, and *Rorc*) and REV-ERBs (*Nr1d1* and *Nr1d2*), which bind to ROR response elements to transcriptionally activate or repress *Bmal1*, respectively. In addition to circadian fluctuation, CLOCK-regulated genes *Nr1d1, Per2*, and *Dbp* fluctuate during the infradian HF growth cycle, being notably upregulated during telogen[5]. Furthermore, mice with mutations in *Clock* and *Bmal1* had delayed anagen progression[5], supporting their role in the regulation of the HF cycle. Emerging evidence suggests that ATRA signaling plays a role in the regulation of biological rhythms; however, currently available data are limited, largely indirect, and contradictory [reviewed in ref. 26].

Recently, we reported that besides the RDH10-DHRS3 retinoid oxidoreductase complex, mouse skin epidermis contains two additional retinol oxidizing enzymes, i.e., retinol dehydrogenase epidermal 2 (RDHE2, SDR16C5) and RDHE2-similar (RDHE2S, SDR16C6). RDHE2 (SDR16C5) and RDHE2S (SDR16C6) together accounted for 80% of microsomal RDH activity in mouse skin at postnatal day (PD) 70. Importantly, a double gene knockout (DKO) of RDHE2 and RDHE2S in mice altered the HF cycle[27]. These mice entered anagen at younger ages than controls. Complete HF cycle analysis was not performed, but preliminary studies indicated high RDHE protein levels in BMP4-positive refractory telogen HFs, but low levels in BMP4-negative competent HFs[28], suggesting that expression levels of RDHEs may fluctuate across the HF cycle.

ATRA concentration in skin is determined through retinol availability, its metabolism to ATRA, and the rate of ATRA degradation; thus, multiple processes must be coordinated to allow for cyclical changes in ATRA levels. This study was undertaken to determine the expression patterns of retinoid metabolic and signaling proteins and ATRA-regulated genes during the entire HF cycle in wild-type (WT) male and female mice *versus* male and female mice lacking the RDHE2 and RDHE2S enzymes. The results of this study uncover a cyclic pattern in ATRA signaling during HF cycle progression, establish a major role of RDHEs in the regulation of ATRA-responsive retinoid metabolic genes and HFSC markers, and provide strong evidence in support of the RDHE's and, hence, ATRA's role in the regulation of molecular clock genes.

## Results
### Loss of RDHE accelerated the hair cycle
Previously, we reported that $Rdhe2^{-/-}/Rdhe2s^{-/-}$ double null mice (DKO) regrew hair faster than wild-type (WT) mice[27]. To better understand hair cycle differences, we collected dorsal skin samples from three-six DKO and WT female mice and three-six DKO and WT male mice across the second synchronized HF cycle at PD25, PD26, PD30, PD35, PD40, PD45, PD48, and PD50, as well as the asynchronous HF cycle at PD54, PD56, PD58, PD60, PD64, and PD68[29] (Fig. 2a). Histological analysis via hematoxylin and eosin staining was used to determine HF cycle stage using the 17-stage guide[29]. Follicles were pooled into early anagen (I–IIIb), mid-anagen (IIIc–IV), late anagen (Anagen V, VI, Catagen I), early catagen (II–V), late catagen (VI–VIII), and telogen (Fig. 2 and Supplementary Fig. 1). We tested normality using Kolmogorov–Smirnov, Shapiro–Wilk, and Levene's tests and then performed either an independent *t*-tests (parametric) or Mann–Whitney *U* tests (non-parametric) at each PD in female and male mice separately to determine genotype differences. During the synchronous hair cycle, female and male DKO mice cycled through anagen and into catagen faster than WT mice. This was only statistically significant for female mice at PD26 (percent of follicles in early anagen, $p = 0.030$, Fig. 2b, c–g) and PD40 (percent of follicles in late anagen, $p = 0.024$, and early catagen, $p = 0.024$; Fig. 2b, h–l). There was more variability in the males. During the asynchronous

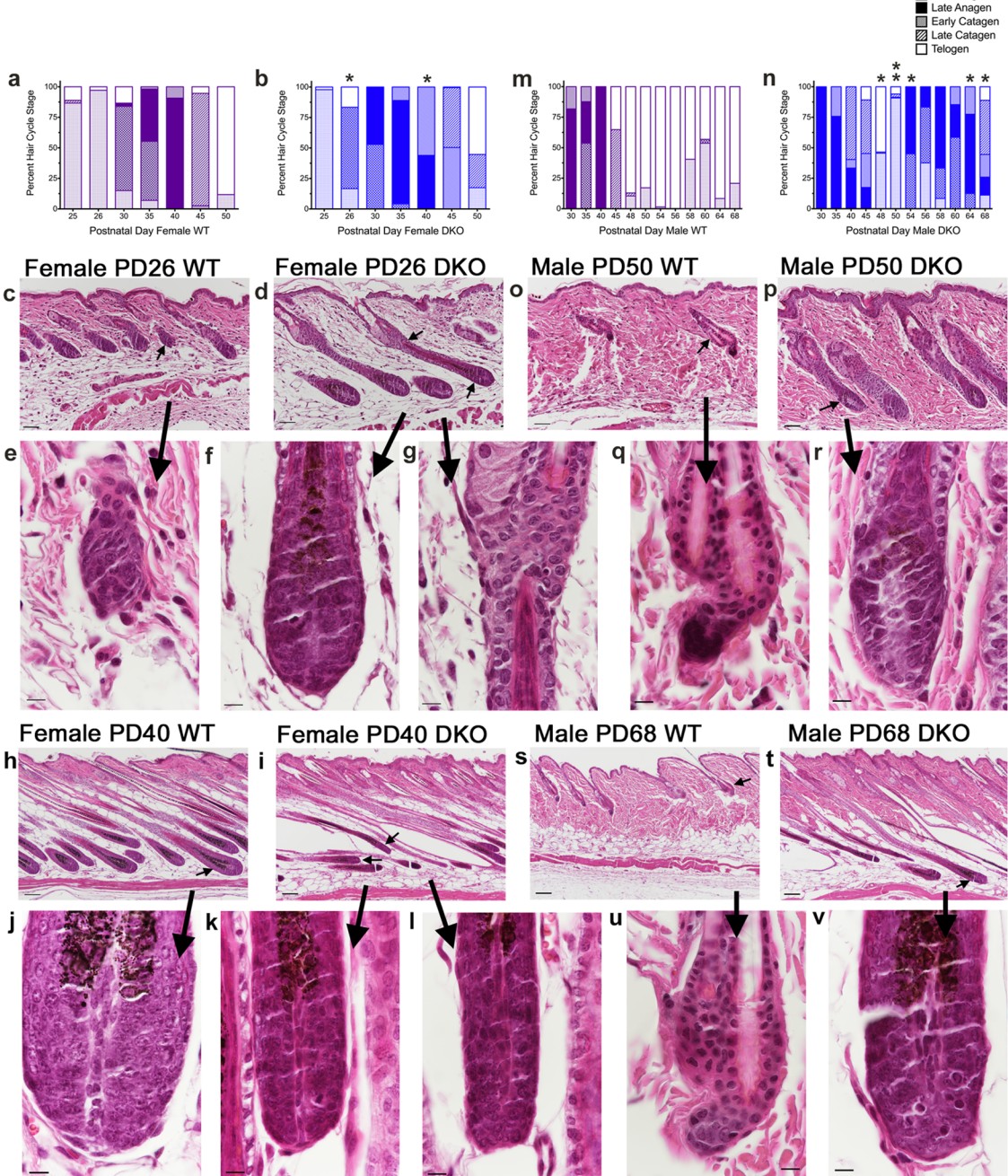

**Fig. 2 | Alterations in the hair cycle in *Rdhe2*⁻/⁻/*Rdhe2s*⁻/⁻ (DKO) mice.** Skin was collected from female (**a–l**) and male (**m–v**) wild-type (WT) (**a, c, e, h, j, m, o, q, s, u**) and *Rdhe2*⁻/⁻/*Rdhe2s*⁻/⁻ null (DKO) mice (**b, d, f, g, i, k, l, n, p, r, t, v**) at postnatal day (PD) 25, 26 (**c–g**), 30, 35, 40 (**h–l**), 45, 50 (**o–r**), 54, 56, 58, 60, 64, and 68 (**s–v**) and stained with hematoxylin and eosin. HF cycle stage was determined using the 17-stage guide[29]. Hair follicles were counted and pooled into early anagen (I–IIIb), mid-anagen (IIIc–IV), late anagen (Anagen V, VI, Catagen I), early catagen (II–V), late catagen (VI–VIII), and telogen. The percent of follicles in each of these stages was then calculated for each mouse. Two to 65 full longitudinal hair follicles were scored per mouse, with two–six mice per group. Normality was tested using Kolmogorov–Smirnov, Shapiro–Wilk, and Levene's tests. Independent *t*-tests (parametric) or Mann–Whitney *U* tests (non-parametric) were then performed at each PD in female and male mice separately to determine genotype differences. *$p < 0.05$ compared to WT for percent early anagen at PD 26, percent late anagen and early catagen at PD 40 in the females; and percent telogen for PD 48, 54, 64, and 68 in males. **$p < 0.005$ compared to WT for percent telogen. Arrows point to the hair follicles magnified below. Bar = 50 μm for (**c, d, o, p**); 10 μm for (**e–g, q, r, j–l, u, v**); and 100 μm for (**h, i, s, t**).

cycle, male DKO mice transitioned from telogen to anagen faster than WT mice, spending less than 5 days in telogen (PD45-50, Fig. 2m–r). Male DKO mice then progressed through anagen, catagen, and into telogen by PD68. In contrast, male WT mice were primarily in telogen from PD48 to PD68, with percent telogen follicles statistically significant at PD48 ($p = 0.034$), PD50 ($p = 0.003$), PD54 ($p < 0.001$), PD64 ($p = 0.003$), and PD68 ($p = 0.006$) (Fig. 2n, s–v).

## RDHE proteins fluctuate across the HF cycle, but the total RDH activity varies in a sex-specific manner

Currently, it is not known whether RDH activity fluctuates during the HF cycle. To better address this question, we used skin from WT mice collected during the synchronous cycle at PD 26 (early anagen), 30 (mid-anagen), 35 (late anagen), 40 (late anagen/catagen), 45 (late catagen), and 50 (telogen). RDH activity is associated with endoplasmic reticulum

membranes[27]; therefore, microsomal fractions were isolated from skin samples for measurements of RDH activity. A Kruskal–Wallis test showed that RDH activity of skin microsomes from female mice varied significantly amongst the PDs ($p = 0.037$), with PD50 significantly decreased 56% from PD30 ($p = 0.043$) (Fig. 3a and Supplementary Table 1). Surprisingly, RDH activity did not vary significantly across PDs in WT male mice (Fig. 3b and Supplementary Table 1). Thus, the total RDH activity of skin microsomes fluctuated across the HF cycle in female, but not male, mice.

Mouse skin microsomes contain three RDHs: RDH10, RDHE2 and RDHE2S[27]. Although two other enzymes, DHRS9 (SDR9C4) and RDH1/16 (SDR9C8) were proposed to catalyze the oxidation of retinol to retinaldehyde[30], recent evidence indicates these two enzymes are more efficient as oxylipin dehydrogenases[31]. We showed that RDHEs account for nearly 80% of microsomal RDH activity from dorsal skin at PD70, during telogen[27]. Whether this is true during other stages of the HF cycle has not yet been established. Therefore, we compared the RDH activity in dorsal skin from WT and RDHE DKO female and male mice at various PDs. For both sexes, a two-way ANOVA showed a main effect of GT ($p < 0.001$), with ~80% decreases in RDH activity in DKO mice across all stages of the HF cycle (Fig. 3c, d and Supplementary Table 1). Thus, RDHEs were the primary RDHs of both male and female skin independent of HF cycle stage, but fluctuation of total RDH activity was observed only in female mice. Importantly, this fluctuation was abolished in skin of RDHE DKO mice (Fig. 3c, blue bars), suggesting the fluctuating pattern of RDH activity was caused specifically by changes in the activity of RDHEs.

Using a custom-made antibody for *Xenopus* Rdhe that detects both mouse RDHE2 and mouse RDHE2S, we assessed RDHE protein levels of female mouse skin microsomes across PD26, 30, 35, 40, and 45 and normalized by cytochrome P450 reductase (POR), a microsome-specific protein (Fig. 3e, Supplementary Fig. 3 and Supplementary Table 1). A Kruskal–Wallis test indicated that both RDHE2 ($p = 0.050$) and RDHE2S

($p = 0.042$) varied across the HF cycle, peaking at PD30 with a subsequent decrease at PD40 for RDHE2 and RDHE2S by 64% for 84%, respectively (Fig. 3e and Supplementary Table 1). Unfortunately, the antibody against POR was discontinued before we collected skin samples at PD50; hence, we were unable to normalize the expression of RDHEs at PD50 per POR. Instead, using the remaining microsomal preparations available for PD45, we established that normalizing RDHE protein levels per total protein amount loaded onto the gel (Fig. 3e, pink bars, Supplementary Fig. 2 and Supplementary Table 1) returned values very similar to those obtained via normalizing per POR (Fig. 3e, compare purple and pink bars for PD45). Protein levels of both RDHEs decreased at PD50 compared to PD45 (Fig. 3e, pink bars), in agreement with the decrease in RDH activity.

Male mouse skin microsomes from all PDs were normalized by total protein as described above for female samples at PD45 and PD50 (Supplementary Fig. 3). A one-way ANOVA showed RDHE2, but not RDHE2S, varied between groups ($p = 0.041$) (Fig. 3f and Supplementary Table 1). However, this variation in RDHE2 protein did not appear to alter the total RDH activity of male microsomes at different HF stages (Fig. 3b, d). RDHE2 is significantly less active than RDHE2S[27]; thus, the contribution of RDHE2 to total RDH activity is likely to be relatively minor and insufficient to significantly alter the total microsomal RDH activity.

In females, both RDHEs' protein levels and RDH activity were elevated at PD30 (mid-anagen), whereas in males RDHEs' levels were elevated at PD45 (late catagen and telogen), but without change in total RDH activity. We also examined *Rdhe2* and *Rdhe2s* transcript levels, anticipating similar variances as seen at the protein level; however, only *Rdhe2* mRNA expression varied across PDs in males ($p = 0.003$) (Fig. 3h and Supplementary Table 1), correlating with changes in protein (Fig. 3f and Supplementary Table 1). *Rdhe2s* did not cycle in males (Fig. 3h and Supplementary Table 1), and neither *Rdhe2* nor *Rdhe2s* transcripts cycled significantly in females (Fig. 3g and Supplementary Table 1). These results suggest that protein levels of RDHE2S in males and females and RDHE2 in females were regulated by post-transcriptional mechanisms.

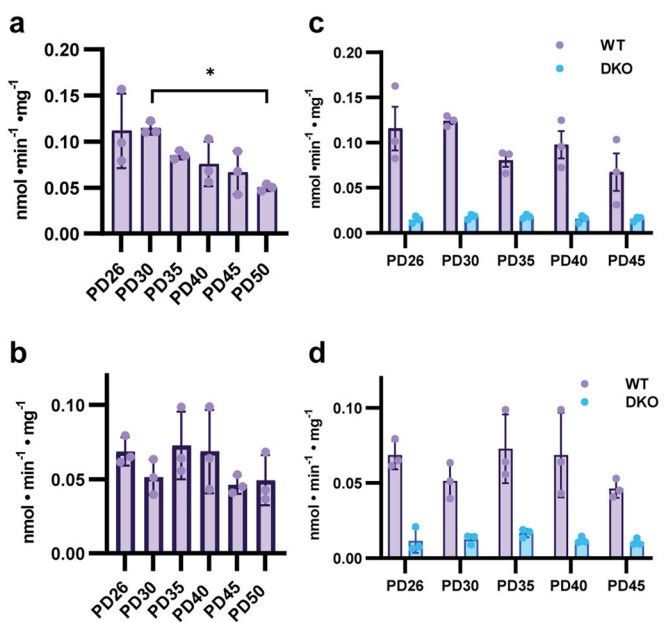
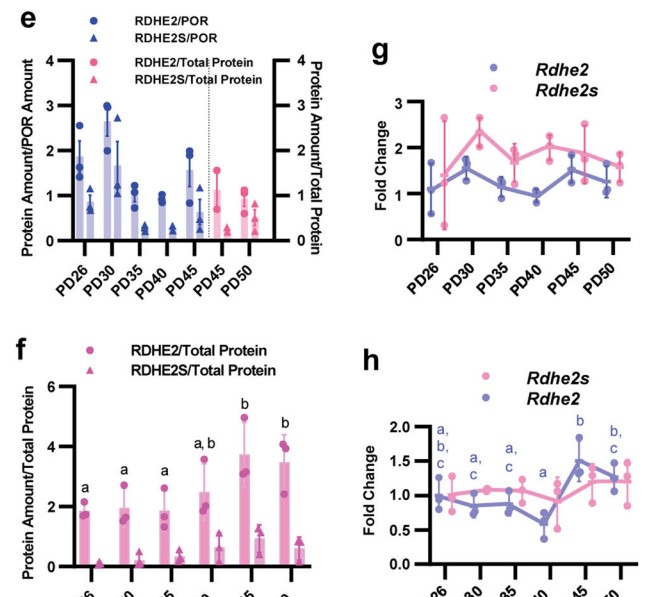

**Fig. 3 | RDHEs fluctuate across the hair cycle, but RDH activity varies in a sex-specific manner. a**, **b** NAD$^+$-dependent retinol dehydrogenase activity of skin microsomal fractions isolated from dorsal skin of WT and DKO female and male mice collected at different PDs ($n = 3$ per PD). **c**, **d** Comparison of RDH activity of microsomes from WT and DKO female and male skin samples at different PDs. **e**, **f** Quantification of RDHE2 and RDHE2S proteins in female and male WT mice by western blot analysis using skin microsomes across PDs. **e** (blue) RDHE2/RDHE2S protein amount was normalized per cytochrome P450 reductase (POR), a microsome-specific marker (**e**), **f** (pink) Due to the discontinuation of POR antibody (Chemicon International, catalog number AB1257) and an inability to find another suitable microsomal-specific antibody, RDHE2/RDHE2S protein amounts were normalized by total protein amount. **g**, **h** *Rdhe2*/*Rdhe2s* mRNA expression across PDs in skin of female and male WT mice was analyzed by qPCR. All data are presented as the mean, with error bars representing SEM. *$p < 0.05$. Letters indicate significant differences, and a Tukey's post hoc was used for (**g**, **h**).

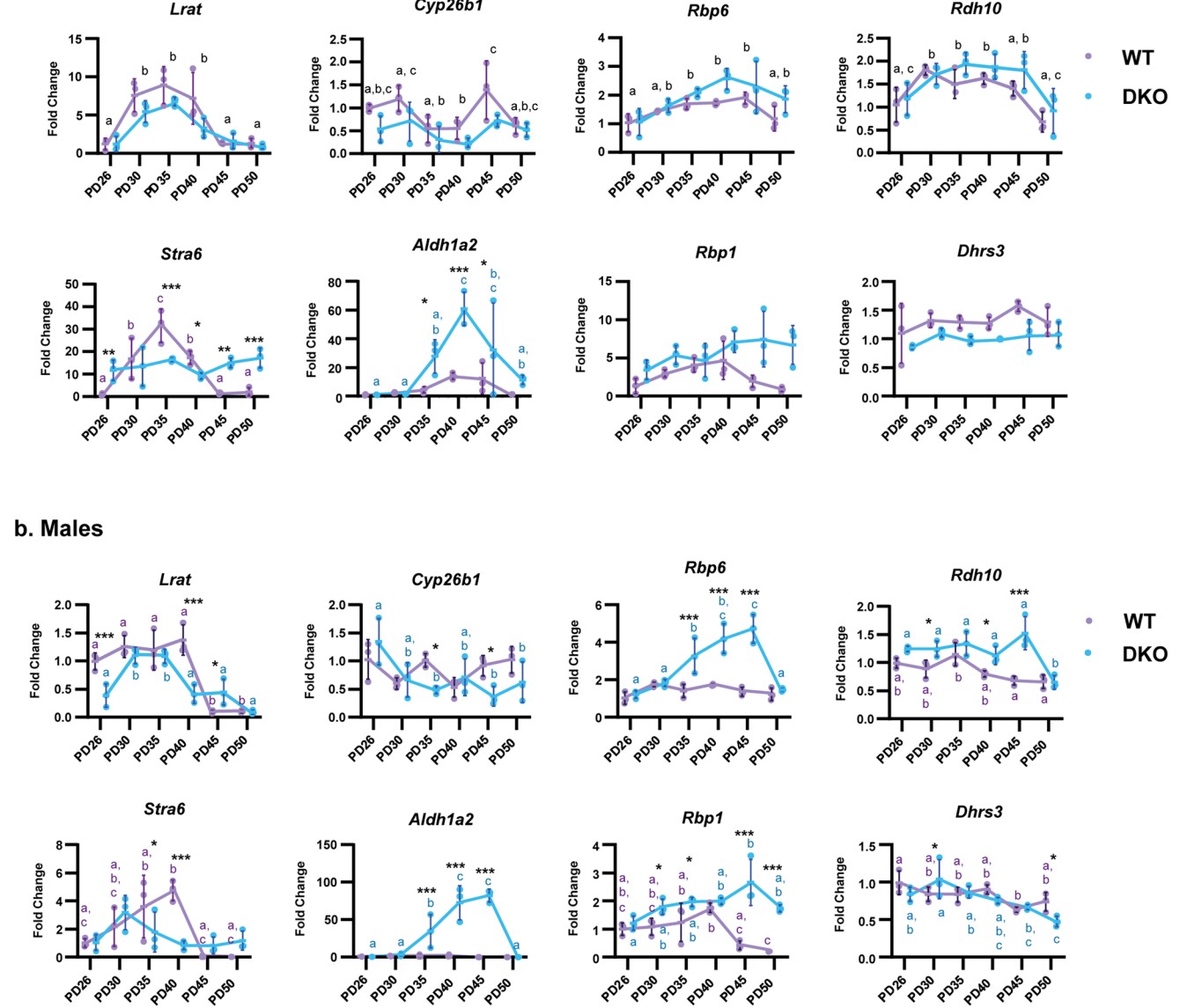

**Fig. 4 | Genetic deletion of RDHE2/RDHE2S alters expression of retinoid metabolic genes and their expression patterns during the hair cycle.** QPCR analysis was performed using the same skin samples taken at PDs outlined in Fig. 3 ($n = 3$ per PD). **a** Female and (**b**) male gene expression patterns. All data are presented as the mean, with error bars representing SEM. All genes were analyzed with a two-way ANOVA. Black letters indicate significant differences by a Tukey's post hoc if there is a main effect of PD and no interaction with genotype. If there is a significant interaction with genotype, asterisks indicate significant differences between genotypes at specific PDs. *$p < 0.05$; **$p < 0.01$; ***$p < 0.001$. Colored letters indicate significant differences between PDs within a genotype according to a Sidak's post hoc. Scatterplots of the data are available in Supplementary Fig. 4.

## ATRA-responsive genes fluctuate across the HF cycle and are dysregulated in skin of $Rdhe2^{-/-}/Rdhe2s^{-/-}$ mice

Several observations suggested that expression of ATRA-responsive genes varies across the HF cycle[15]. However, whether they show cyclic patterns is not known. To address this question and to determine whether genetic deletion of RDHEs affects these patterns, we performed qPCR analysis of well-known markers of ATRA signaling using RNA prepared from the same samples of WT and DKO skin used for activity assays and western blotting. QPCR analysis showed that in WT female mice, expression of *Lrat* ($p < 0.001$), *Cyp26b1* ($p = 0.003$), *Rbp6* ($p = 0.001$), and *Rdh10* ($p < 0.001$) fluctuated across the HF cycle (Fig. 4a, Supplementary Table 2 and Supplementary Fig. 4). Notably, in female DKO skin, the amplitude of fluctuations in *Lrat*, *Cyp26b1*, and *Rbp6* was reduced compared to female WT skin. DKO female mice also displayed lower total levels of *Lrat* ($p = 0.013$), *Cyp26b1* ($p = 0.001$), and *Dhrs3* ($p < 0.001$) transcripts across all PDs. *Stra6*

expression varied across PDs in WT female mice ($p < 0.001$), exhibiting an interaction with PD and GT ($p < 0.001$), and this variance was nearly abolished in DKO mice.

Surprisingly, expression of *Aldh1a2* across PDs varied in DKO female mice ($p < 0.001$) but not in WT female mice, with an interaction of PD and GT ($p = 0.010$). The increase in *Aldh1a2* transcript levels reached 30–60-fold in DKO female mice at PD35 ($p = 0.018$), PD40 ($p < 0.001$), and PD45 ($p = 0.041$). *Rbp1* did not fluctuate across PDs in either WT or DKO skin but was generally upregulated in DKO female mice ($p < 0.001$). A mild upregulation of *Rbp6* ($p = 0.004$) and *Rdh10* ($p = 0.027$) was also observed in DKO female mice.

Male mice displayed notably different patterns of gene cycling, with many of the genes having significant interactions of PD and GT. *Lrat* cycled in both GTs and showed a GTxPD interaction ($p < 0.001$) (Fig. 4b and Supplementary Table 2), where expression in DKO skin was significantly

different from WT skin at PD26 and PD40 ($p = 0.001$, $p < 0.001$). Similar to female mice, $Stra6$ had an interaction ($p = 0.001$) where its expression varied across PDs only in WT mice ($p < 0.001$). Some interactions appeared as small alterations to expression patterns, as in the GTxPD interactions of $Cyp26b1$ ($p = 0.023$) and $Dhrs3$ ($p = 0.027$). However, $Rbp1$ ($p < 0.001$) and $Rdh10$ ($p = 0.003$) were significantly upregulated for most PDs in DKO male mice. A highly significant interaction between PD and GT for both $Rbp6$ and $Aldh1a2$ ($p < 0.001$) resulted in variance across PDs only in DKO ($p < 0.001$) but not WT male mice, with significant upregulation in DKO male mice at PD35, PD40, PD45 ($p < 0.001$) (Fig. 4b and Supplementary Table 2).

Expression of $Rbp1$, $Rbp6$, $Aldh1a2$, and $Rdh10$ was increased in DKO mice of both sexes, suggesting potential activation of compensatory mechanisms due to the lack of RDHEs. Genes generally upregulated by ATRA ($Stra6$, $Lrat$, $Dhrs3$, and $Cyp26b1$) exhibited dysregulated expression patterns, though there were no consistent patterns across both sexes, except for the downregulation of $Lrat$.

## Expression of HFSC markers is diminished in skin of $Rdhe2^{-/-}/Rdhe2s^{-/-}$ mice

In our previous study, we found expression levels of several HFSC markers altered at PD70 in skin of DKO mice[27]. To better understand the relationship between cyclic ATRA signaling and hair growth, we performed qPCR for prominent HFSC markers with known expression in the bulge region: $Sox9$, $CD34$, $Krt15$, and $Lgr5$[32–39]. Our data showed that all four HFSC markers significantly fluctuated across the HF cycle in WT mice (Fig. 5, Supplementary Table 3 and Supplementary Fig. 4). Importantly, HFSC markers were decreased in DKO animals across the majority of the HF cycle stages. In females, transcripts for all four genes were significantly lower in DKO mice ($Cd34$ $p = 0.001$; $Sox9$, $Krt15$ $p < 0.001$; $Lgr5$ $p = 0.002$). $Lgr5$ expression was affected by both GT and PDs ($p = 0.012$), with a complete loss of PD variance in DKO mice (Fig. 5a). In male mice (Fig. 5b), we observed an interaction of GT and PD for $Cd34$ ($p = 0.018$) and $Krt15$ ($p < 0.001$), as DKO mice did not exhibit the same increases in transcripts for $Cd34$ and $Krt15$ at PD45 and PD50 that were observed in WT mice. $Sox9$ expression showed an interaction of PD and GT ($p = 0.024$), with a unique pattern for WT and DKO mice. In both GTs, $Sox9$ expression decreased significantly at PD50. $Lgr5$ did vary across PDs ($p < 0.001$), but there was a notable absence of peaks at PD30 and PD45 in female DKO mice and at PD35 in male DKO mice. In both sexes, $Cd34$ and $Krt15$ demonstrated similar expression patterns, with increases in mRNA at late catagen/telogen, while $Sox9$ and $Lgr5$ appeared to be more highly expressed in anagen. IHC was performed to compare expression in stage matched HFs (Fig. 5c). Consistent with qPCR, CD34 was greater in telogen HFs from WT mice in both sexes. Krt15 expression was more dependent on precise HF cycle stage and was too variable to confirm a difference. SOX9 was likely greater via qPCR in male DKO mice at PD45 due to differences in HF cycle. SOX9 localized to the outer root sheath, bulge, and differentiated sebocytes. There was no difference in percent of SOX9 positive HFs or immunoreactivity, but DKO mice have longer HFs with more outer root sheath cells. At PD45, percent SOX9 positive sebocytes was lower in DKO mice, however this could also be stage related. In agreement with our data, $Krt15$ and $Cd34$ were shown to be more prominently expressed in telogen bulge cells[32,40]. Thus, deletion of RDHEs significantly altered the fluctuation pattern and the overall expression levels of HFSC marker CD34 in a sex-specific manner.

## Awl hairs are increased in skin of $Rdhe2^{-/-}/Rdhe2s^{-/-}$ mice

DKO mice had a noticeably disheveled hair coat compared to WT littermates (Fig. 6a, b and Supplementary Table 4), though there were no histological abnormalities in the HF[27]. Previous studies suggested that altered ATRA signaling can affect HF development[41]. Mouse dorsal skin contains four distinct hair types (guard, awl, auchene and zigzag), distinguished based on the length of the hair shaft, the number of medulla cells (thickness), and the presence of kinks (bends) in the shaft. Guard hairs make up ~1% of the back skin hairs and are the longest with no kinks[42]. Awl hairs are straight

but shorter and thicker than guard hairs. Auchene hairs are identical to awl hairs with one kink in the hair shaft[43,44]. Together, awl/auchene make up ~17% of back hair. Finally, zigzag hairs are the most abundant (~81% of hairs) and the thinnest hairs characterized by at least two bends in the hair shaft, giving them a characteristic "z" shape.

Increased ATRA levels in back skin of $Cyp26b1^{-/-}$ mice were shown to inhibit bending of hair fibers and decrease the percentage of zigzag hairs[41]. To assess hair fiber composition in DKO mice, over 100 hairs per mouse were collected from the middle back and categorized into guard, awl, auchene, and zigzag hair[44,45]. Interestingly, DKO mice had a 1.6-fold higher percentage of awl hairs compared to WT littermates ($p = 0.008$), without a corresponding decrease in any other hair fiber type (Fig. 6c). Thus, the reduced ATRA signaling in skin of DKO mice clearly affected the hair follicle morphogenesis but produced a different outcome than excessive ATRA levels in skin of CYP26B1 KO mice, which inhibited bending of hair fibers.

## Expression levels of molecular clock genes and Notch1 are reduced in skin of $Rdhe2^{-/-}/Rdhe2s^{-/-}$ mice

As reported previously, DKO mice entered anagen at a younger age than controls[27]. To determine whether the altered HF cycle in DKO mice could be related to changes in molecular clock genes, we examined the expression levels of six CLOCK-regulated genes ($Dbp$, $Npas2$, $Nr1d1$, $Per2$, $Cry2$, and $Clock$) at PD50, when an upregulation of $Dbp$, $Nr1d1$, and $Per2$ normally occurs in WT mouse skin[5]. QPCR analysis showed that expression of $Dbp$, $Npas2$ (paralog of $Clock$), and $Nr1d1$ was significantly reduced in female DKO mice compared to WT female littermates (Fig. 7a and Supplementary Table 5). In male DKO mice, transcripts for four genes ($Dbp$, $Npas2$, $Per2$, and $Cry2$) were lower (Fig. 7b), suggesting dysregulation of molecular clock genes.

Progression into anagen is also regulated by cross-talk of the Wnt/β-catenin, Hedgehog, and Notch signaling pathways [reviewed in refs. 46,47]. Activation of Wnt/β-catenin signaling is required for HF regeneration whereas Notch1 antagonizes β-catenin activity. The first HF cycle of $Notch1$ deficient mice is characterized by shortened anagen and a premature entry into catagen[48]. Since we saw an earlier entry into catagen in DKO mice, we analyzed expression pattern of $Notch1$ across the HF cycle in DKO versus WT female and male mice. $Notch1$ was significantly different across the PDs in both males ($p = 0.002$) and females ($p < 0.001$), but only females showed a difference in GT ($p = 0.043$)—DKOs had overall decreased $Notch1$ compared to WTs (Fig. 7c). Thus, the decreased expression of $Notch1$ could contribute to the earlier catagen entry in females.

## Discussion

To our knowledge, this study represents the first investigation of ATRA-regulated genes, HFSC markers, and clock genes across all stages of the HF cycle in female and male mice. Although previous studies suggested ATRA levels might vary during the HF cycle, direct evidence was lacking. ATRA concentration in skin is very low (≤20 nM)[49]. Thus, it would be challenging or nearly impossible to detect variations in ATRA skin concentrations at different stages; however, the cumulative analysis of multiple ATRA target genes as readout of ATRA signaling strongly suggests that ATRA signaling occurs in a cyclic pattern. This conclusion is further supported by the finding that RDHE2 and RDHE2S, the major enzymes catalyzing the rate-limiting step of ATRA biosynthesis at all HF stages, also display a cyclic pattern.

Interestingly, although RDHEs proteins cycle in both males and females, the pattern is different. In females, both RDHE2 and RDHE2S drop at PD50; but in males, RDHE2S does not change significantly while RDHE2 rises at PD45 and PD50. Despite this rise in RDHE2 in males, the overall RDH activity in male skin does not increase significantly, because RDHE2 is significantly less active than mouse RDHE2S and, hence, might not have a big impact on overall RDH activity. In females, the decrease in the total RDH activity correlates well with the decrease in RDHE2S and RDHE2 protein levels.

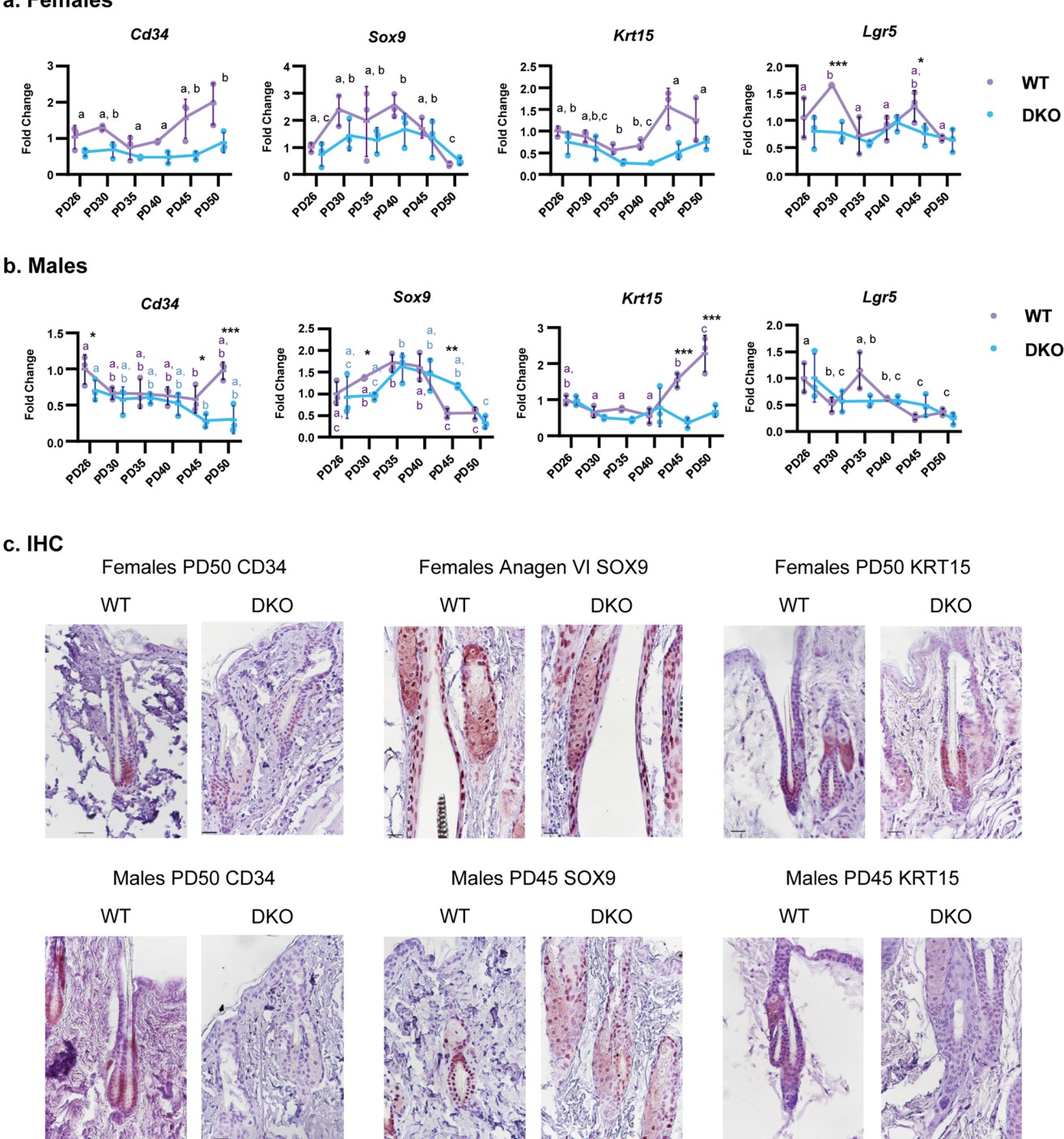

**Fig. 5 | Expression of hair follicle stem cell markers is diminished and the cycling pattern is altered in RDHE DKO mice.** Using the same samples from Fig. 2, qPCR was performed for genes identified in the literature as markers of hair follicle stem cells in the bulge of the hair follicle ($n = 3$ per PD). **a** Females, **b** males. All data are presented as the mean, with error bars representing SEM. All genes were analyzed with a two-way ANOVA. Black letters indicate significant differences by a Tukey's post hoc if there is a main effect of PD and no interaction with genotype. If there is a significant interaction with genotype, asterisks indicate significant differences between genotypes at specific PDs. *$p < 0.05$; **$p < 0.01$; ***$p < 0.001$. Colored letters indicate significant differences between PDs within a genotype according to a Sidak's post hoc. Scatterplots of the data are available in Supplementary Fig. 4. **c** Immunohistochemistry of representative hair follicles for stem cells markers using a red chromogen. Bar = 25 μm.

In either sex, variation of RDH activity or RDHEs' protein levels is not related to changes in size of the HF, i.e., a bigger HF does not contain more RDHE protein. For instance, in females, RDH activity is the highest at PD30 (mid-anagen); however, the HF reaches its maximum size at PD35 and PD40 (late anagen/catagen). Males exhibit their highest RDHEs' protein levels at PD45 (late catagen), when the HF is rapidly decreasing in size. Thus,

the observed variability in RDHE proteins and RDH activity reflects a well-orchestrated developmental program rather than simple changes in HF size.

Other retinoid metabolic and signaling proteins were also shown to be differentially expressed in males and females[50]. Immunohistochemical analysis demonstrated that skin with hair follicles in mid-anagen of female mice had elevated levels of ALDH1A2, ALDH1A3, and CRABP2, compared

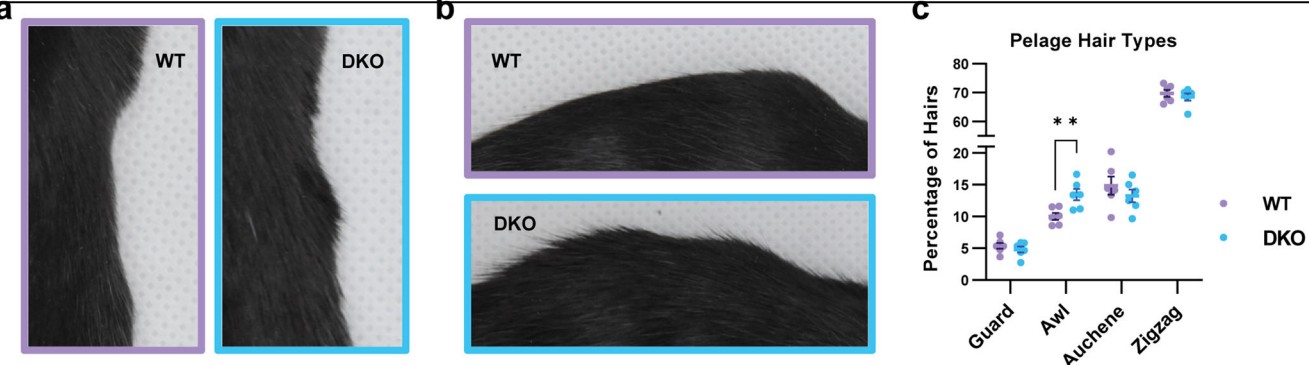

**Fig. 6 | Awl hairs are increased in male and female RDHE DKO mice.** Images of the (**a**) side and (**b**) back of WT and DKO mice in a lightbox. **c** Quantification of pelage hair types, shaved from the middle back of mice from PD58–PD61 (during telogen, confirmed via identification of pink skin once shaven). At least 100 hairs were counted per mouse and categorized under a dissection microscope, $n = 3$ males, 3 female; **$p < 0.01$.

to males, which had higher levels of ALDH1A1 and RARβ. Androgens and estrogens are known modulators of the HF cycle[51], and their dysregulation is capable of causing alopecia. Additionally, androgens and estrogens interact with retinoids. Androgen signaling is inhibited by ATRA in multiple tissues[52–54]. However, estrogen signaling appears to have a cooperative interaction with RARα[50]. Estrogen receptors (α and β) are localized to the HF primarily during anagen, which coincides with elevated RDH activity and higher RDHEs' expression in females.

In addition to establishing the cyclic nature of ATRA signaling, we discovered interesting gene-specific patterns of retinoid metabolic gene expression. For example, expression pattern of *Cyp26b1* closely followed the expression pattern of RDHEs, with peaks at PD30 and PD45, suggesting the increased activity of RDHEs generated an increase in ATRA that, in turn, induced expression of *Cyp26b1*. *Stra6* and *Lrat* were coordinately upregulated at PD35 in females and at PD40 in males, suggesting that increased flux through STRA6 was tied to accumulation of retinyl esters by LRAT. The mRNA expression patterns observed in this study are supported by the immunolocalization analysis available for some retinoid proteins[16]. For example, similarly to mRNA expression, ALDH1A2 protein was detected in the cycling HF from mid to late anagen (anagen IIIc) through catagen in females. Likewise, both RBP6 mRNA and protein appeared in late anagen and peaked in anagen VI/catagen I before fading[16].

RDHEs clearly have a major role in the regulation of retinoid homeostasis in skin. Genetic deletion of RDHEs not only altered the overall abundance of ATRA-regulated transcripts, but disrupted their fluctuation patterns. Interestingly, there was an apparent dichotomy in the response of several genes. Some genes (*Rbp1, Rdh10, Aldh1a2, Crabp2*) were upregulated in RDHE DKO mice compared to WT mice, while other genes (*Lrat, Dhrs3, Cyp26b1*) exhibited decreased expression levels in DKO mice. Since *Lrat, Dhrs3,* and *Cyp26b1* strongly correlate with ATRA signaling, we infer that the decrease in their expression levels is indicative of reduced ATRA signaling in skin. The increase in transcript levels for *Rbp1, Rbp6, Aldh1a2,* and *Rdh10* in DKO mice of both sexes could be due to activation of compensatory mechanisms to maintain ATRA levels.

RDHE DKO mice have notably disheveled back hairs that do not lie smoothly like the coat of WT littermates. The hair phenotypes in genetically-altered mouse models are often unreported and thus, little is known about the mechanisms defining the hair composition. We found that DKO mice have an increased percentage of awl hairs, which are induced in the second wave of HF growth at E16. Awl hairs are straight and the second longest hair type. They are the thickest type of hair possessing up to 4 rows of medulla cells within the hair cuticle[55]. Alongside awl hairs, auchene hairs contain 4 rows of medulla cells, but exhibit a characteristic bend. Auchene hairs are also induced in the second wave of HF induction, and some studies group these two hair types. However, our results show no differences in auchene hair percentages, indicating RDHEs are relevant to developmental programming defining the divergence of two hair types during the second wave of hair induction.

Elevated levels of ATRA were shown to restrict bending of the hair fiber and development of zigzag hairs[41]. The lack of changes in bending or percentage of zigzag hairs further indicates that DKO mouse skin has decreased rather than increased ATRA levels, and that reduced ATRA increases the percentage of HF that produce awl hairs.

The altered hair coat composition and accelerated entry into anagen suggested that ATRA produced as a result of RDHE activity regulates HFSC maintenance, quiescence, and differentiation. Indeed, transcript and protein levels for CD34 were diminished in DKO mice, indicating that precise ATRA levels regulated by RDHEs are important not only for generation of specific hair types but also for maintaining HFSCs. The quick 5-day transition from catagen to early anagen (PD45-50) in DKO male mice also suggests that ATRA is important for maintaining HFSC quiescence in telogen. This second telogen typically takes 35 days before anagen is initiated[29]. This loss of quiescence depletes HFSCs[56], which we saw by PD50 with the reduction of CD34. If the accelerated HF cycle continues, it could lead to early aging, as HFSCs would continue to be lost[56]. The few available reports on the effects of ATRA and its synthetic analogs on the HF cycle have been contradictory[57–60], likely due to dose and/or time with a possible U-shaped curve. Excess endogenous ATRA prolonged anagen and shortened the second telogen in *Dgat1*−/− null mice, which is involved in retinyl ester synthesis[57]. In contrast, high dose pharmacological oral retinoids, such as acitretin and etretinate, arrested human HFs in telogen[59,60]. Yet treatment of cultured anagen HFs with ATRA reduced anagen and triggered catagen[58]. Dietary studies in mice also found variable results on the HF cycle due to dose and the timing of diet changes[15,61]. These studies were performed before RDHEs were found to play a major role in skin retinoid homeostasis; thus, the effects of retinoid treatments on RDHE activity have not been considered. Notably, human RDHE2 (SDR16C5), the single ortholog of mouse RDHEs in humans[62], is downregulated in human keratinocytes by elevated ATRA signaling[63], suggesting a negative feedback regulatory loop.

Infradian rhythms are known to be regulated by seasonal changes in light conditions, ambient temperature and food availability. These seasonal changes affect the length of hair in specific body sites and regulate the periodic shedding of fur in mammals. Vitamin A availability clearly affects the HF cycle because increasing the vitamin A in the diet raises the percent of HFs in anagen[28,61]. The results of this study suggest that vitamin A and ATRA signaling regulates the HF cycle not only through the HFSCs, but also by modulating the molecular clock genes. Expression of *Npas2* and *Dbp* was significantly lower during telogen in both female and male skin, and there were also sex-specific differences in expression of *Nr1d1* (lower in females) and *Per2* and *Cry2* (lower in males). It should be noted that the analysis of molecular clock genes was carried out at a singular time point during the day. To determine whether the observed differences were due to an overall decrease in expression of clock genes across the 24-h cycle or an altered oscillation pattern of clock genes, we would need to collect skin samples at multiple points across the 24-h cycle. This study will be conducted in the

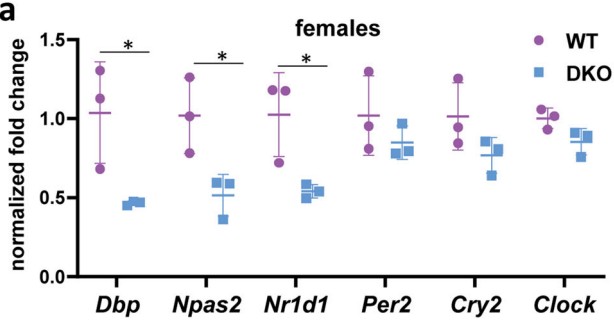

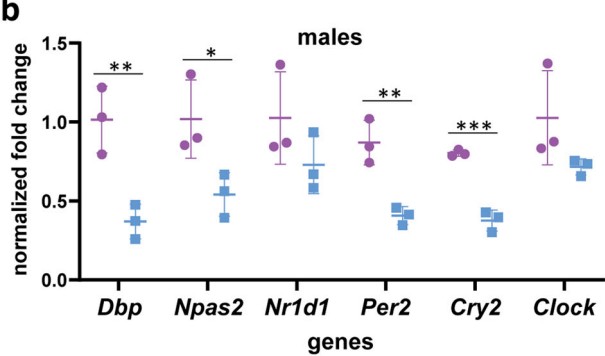

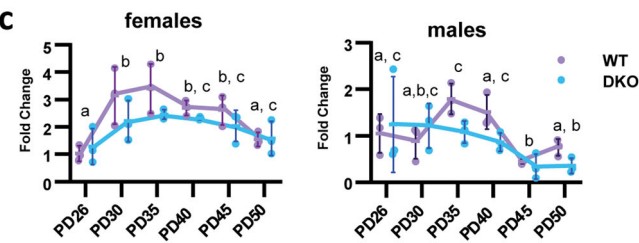

**Fig. 7 | Expression of several clock genes is reduced in male and female RDHE DKO mice and the cycling pattern of Notch1 is altered.** Using samples from PD50 (n = 3), qPCR was performed for molecular clock genes for (**a**) females and (**b**) males. In (**c**), *Notch1* was examined across all PDs due to its importance in hair cycle progression. All data are presented as the mean, with error bars representing SEM. Clock genes at PD50 were analyzed via a *t*-test and *Notch1* was analyzed with a two-way ANOVA. Black letters indicate significant differences by a Tukey's post hoc for a main effect of PD. Asterisks indicate significant differences between genotypes. *$p < 0.05$; **$p < 0.01$; ***$p < 0.001$.

future. Nonetheless, the altered expression of clock genes demonstrates the role of ATRA signaling in modulating biological rhythms.

In summary, this study uncovers the essential role of RDHEs in cyclic regulation of skin retinoid homeostasis, hair type composition, and HF cycle progression, providing insights into our understanding of the molecular mechanisms governing the HF regeneration. Current findings should inform the design and interpretation of future studies investigating the effects of retinoids and rexinoids on HF cycle and HFSCs in human and mouse skin.

## Methods

### Mice
*Rdhe2/Rdhe2s* double knockout mice were generated in previous studies via CRISPR technology on a C57BL/6J background[27]. *Rdhe2*$^{+/-}$;*Rdhe2s*$^{+/-}$ mice were bred to obtain WT and *Rdhe2*$^{-/-}$;*Rdhe2s*$^{-/-}$ littermates for these studies. Genotyping for *Rdhe2/Rdhe2s* was carried out using primers Sdr16c5ex5F, 5'-CAG ACT ATT GTG CAA GTA AAT TCG C-3', and Sdr16c5ex5R, 5'-TGG GCA GAG AGT AAA TTT GAA TGC C-3', to identify the WT allele and Sdr16c6ex5F, 5'-ATA CTC TGT CCT CAA GGA

TAA ACC-3', and Sdr16c5ex5R or Sdr16c5intr4F4, 5'-CTT GAG ATA ATC AAC TTG AAA GGA G-3', and Sdr16c5intr5R2, 5'-GAA TGG GTC TGA ATG GCA TTA CG-3', to identify the DKO deletion of *Rdhe2/Rdhe2s*. Mice were housed at 23 ± 2 °C with free access to water and food (standard rodent chow diet obtained from Harlan, catalog number 7017) in an AALAC-approved pathogen-free facility on a light cycle of 12 h light and 12 h dark. All studies were conducted with approval of the Institutional Animal Care and Use Committee of the University of Alabama at Birmingham School of Medicine. We have complied with all relevant ethical regulations for animal use.

Male and female mice were euthanized on postnatal days 26, 30, 35, 40, 45, 48, 50, 54, 56, 58, 60, 64, and 68 using $CO_2$. The skin taken from an individual mouse was used for multiple assays at one time point and not multiple time points. The wound healing processes were not triggered because all skin was taken right after sacrifice. Dorsal skin was shaved and collected with a small piece fixed overnight in 10% Buffered Formalin Phosphate, transferred to 70% ethanol, processed routinely, embedded in paraffin, sectioned at 10 μm, placed on microscope slides (Superfrost Plus Fisherbrand, Pittsburgh, PA), and stained with hematoxylin and eosin (H&E) for histological analysis. HF cycle stage was determined using the 17-stage guide[29]. Hair Follicles were counted and pooled into early anagen (I–IIIb), mid-anagen (IIIc–IV), late anagen (Anagen V, VI, Catagen I), early catagen (II–V), late catagen (VI–VIII), and telogen). The percent of follicles in each of these stages was then calculated for each mouse. Two to 65 full longitudinal hair follicles were scored per mouse, two–six mice per group. The rest of the skin was cleaned of the underlying connective tissue and stored at −80 °C for biochemical analysis. For hair fiber collection, mice were sedated with isoflurane and hair was shaved and collected from the middle back. Hair types were categorized under a dissection microscope by number of medulla cell rows and number of bends[44,45].

### Isolation of microsomal fractions from skin
Microsomes and mitochondria were isolated by differential centrifugation in a sucrose gradient[27]. Approximately 100 mg of frozen skin tissue samples were homogenized on ice by Polytron (Biospec Products, Inc., Model 985370) in 3 bursts for 10 s each in 1 ml of ice-cold Isolation buffer (0.25 M sucrose in PBS supplemented in 1 mM EDTA and protease inhibitors: 1 μg/ml aprotinin, 1 μg/ml leupeptin, and 1 μg/ml pepstatin A). Crude homogenates were further homogenized using 20 strokes in a glass homogenizer on ice. Samples were centrifuged at 3000 × g for 10 min at 4 °C. Supernatant was removed into pre-labeled 2 ml tubes for mitochondrial isolation; pellets were washed with 300 μl of buffer and re-centrifuged at 3000 × g for 5 min. The supernatant was also removed into the same pre-labeled 2 ml tube for mitochondrial isolation. The 3000 × g supernatants were centrifuged at 10,000 × g for 10 min at 4 °C. The supernatant was removed into 3 ml conical ultra-centrifuge tubes and balanced with isolation buffer before being centrifuged at 105,000 × g for 90 min at 4 °C. The 10,000 × g pellet is the mitochondrial fraction and was re-suspended in 100 μl reaction buffer (90 mM K$_2$HPO$_4$/KH$_2$PO$_4$, 40 mM KCL) supplemented with 20% glycerol and 1 mM EDTA. After ultra-centrifugation, the supernatant was removed and the microsomal pellet was washed with 100 μl of isolation buffer, making sure not to disturb the pellet. The wash was discarded and the pellet re-suspended in 100 μl of reaction buffer supplemented with 20% glycerol and 1 mM EDTA before being transferred to pre-labeled 1.5 ml Eppendorf tubes. All collected samples were flash frozen after re-suspension and stored at −80 °C.

### Western blot analysis
Protein concentrations were determined according to Peterson's modification with BSA as a standard[64]. Thirty μg of microsomal fractions were loaded on SDS-PAGE using standard Laemmli system with 12% separating gel and 4.5% stacking gel (mini gel, 160 V). A PVDF membrane was pre-soaked in ethanol and gel transfer occurred for 75 min using Semi Dry transfer unit. Blots were incubated with Ponceau S solution (0.1% Ponceau, 5% acetic

acid) and imaged (BIO-RAD, ChemiDoc™ MP Imaging System) for total protein quantification. Blots were blocked in 5% BSA in TBST for 1 h.

Blots were probed overnight in a cold room with custom-made *Xenopus* rdhe2 antibodies at 1:4000 dilution and rabbit Cytochrome P450 Reductase antibodies diluted 1:4000 (Chemicon International, catalog number AB1257) in 5% BSA in TBST. Secondary ECL Plex goat anti-rabbit IgG, Cy®5 (Sigma, catalog number PA45011) was used at a 1:2500 dilution in 5% BSA in TBST. Blots were scanned by an Amersham Typhoon to visualize fluorescent bands.

### Retinol dehydrogenase activity in microsomal skin fractions

Fifty µg of microsomal skin fractions was incubated with 3 µM all-*trans*-retinol (Toronto Research Chemicals, Toronto, Canada) solubilized with bovine serum albumin and 1 mM $NAD^+$ (Sigma-Aldrich, St. Louis, MO) in 0.5 ml of reaction buffer for 15 min at 37 °C[65]. Reactions were stopped by the addition of an equal volume of ice-cold methanol and retinoids were extracted with 2 ml of hexane. Hexane layers were dried, and dry residue was reconstituted in 0.1 ml of hexane:ethyl acetate (90:10). Retinoids were separated by normal-phase HPLC using a Spherisorb S3W column (4.6 mm × 100 mm; Waters, Milford, MA) and an isocratic mobile phase consisting of hexane:ethyl acetate (90:10) at 1 ml/min[65].

### Quantification of gene expression in skin samples

Approximately 100 mg of frozen dorsal skin tissue of WT and DKO mice samples was homogenized in PURzol reagent (Bio-Rad, catalog no. 7326890) and RNA isolated via the Aurum™ Total RNA Fatty and Fibrous Tissue Pack (Bio-Rad, catalog no. 732-6870). The concentration of isolated RNA was determined with a Nanodrop ND-1000 spectrophotometer (Thermo Scientific). ProtoScript® II First Strand cDNA Synthesis Kit (New England BioLabs, catalog no. E6560) was used for cDNA strand synthesis from 1 µg sample RNA. One WT mouse sample was used for several extra reactions simultaneously as a standard for quantification curves. All qPCR was performed with a 15X dilution of cDNA, unless otherwise noted. Curves were calculated with 3X, 9X, 27X, and 81X cDNA dilutions. Real-time PCR analysis was conducted on a Roche LightCycler®480 detection system (Roche Applied Science) with SYBR Green as the probe (LightCycler®480 CYBR Green I Master, Roche Applied Science). Gene expressions were normalized to *Gapdh* and analyzed as a relative expression of fold-difference from the expression level of WT mice at PD26 of the same sex, using the comparative *Ct* method. Sequences of primers are available by request.

### Immunohistochemistry (IHC)

Formalin-fixed Paraffin-Embedded (FFPE) tissue samples were de-waxed and rehydrated using Xylene and decreasing concentrations of ethanol. The following antibodies and dilutions were used in this study: CD34 (rat, 1:25; BD Pharmingen), SOX9 (rabbit, 1:10,000; abcam), KRT15 (rabbit, 1:40,000; abcam). For tissue samples targeting antibodies (Abs) CD34 and SOX9, antigen unmasking was performed using a heat-mediated Citrate buffer (pH 6.0). All tissue samples were treated with 3% hydrogen peroxide, blocked with 3% bovine serum albumin (BSA) plus 2% normal goat serum (NGS; Jackson ImmunoResearch Laboratories, Inc., West Grove, PA.) and Streptavidin and Biotin blocking kit (Vector Laboratories, Newark, CA). Samples were incubated overnight with primary antibodies at 4 °C. To amplify the signal, biotinylated anti-rabbit or anti-rat (CD34) secondary antibodies (1:500, Jackson ImmunoResearch Laboratories, Inc., West Grove, PA.) and then horseradish per-oxidase conjugated (HRP) anti-biotin tertiary Abs (1:200; Bethyl Laboratories, Montgomery, TX) were applied for 1 h each at room temperature. Vector Laboratories NovaRED® (Newark, CA) substrate kit was utilized followed by counterstaining with Gils Hematoxylin III (Poly Scientific, Bay Shore, NY). Tissue samples were dehydrated using increasing concentrations of ethanol and Xylene and mounted with a non-aqueous mounting serum (Vector Laboratories, Newark, CA) and coverslips. Hydrogen peroxide and BSA were procured from Fisher

Scientific (Hampton, NH). IHC and H&E images were captured on a Nikon Eclipse i80 microscope with the DS-Ri2 camera (Nikon Instruments Inc, Melville, NY). These images were minimally sized, arranged, and brightness and contrasted adjusted to the whole image similarly for all treatments using Photoshop Elements 15 (Adobe). Immunoreactivity (IR) was scored on a four-point scale (0–4) and hair follicles and sebocytes with each IR score counted. Percent positive HF and sebocytes were calculated as the sum of IR 3 and 4 counts divided by the total number of HFs or sebocytes times 100.

### Statistics and reproducibility

Males and females were analyzed separately because of their differences in HF cycle progression. Statistical significance was determined using one-way ANOVA for RDH activity and for Western blot comparison of RDHE2 and RDHE2S across the HF cycle, followed by a Tukey test post hoc analysis. When parametric assumptions were not met, a Kruskal–Wallis was performed instead. For hair cycle stage, a two-way ANOVA was performed for PD and GT. When interactions were seen, this was followed by two-tailed *t*-tests or Mann–Whitney *U* tests (non-parametric) between genotypes at each PD. For qPCR, a two-way ANOVA was used to determine main effects of PD and GT and possible interactions, with a Sidak test for estimated marginal means. Two-tailed *t*-tests were used to compare WT and DKO mice pelage hair percentages and PD50 genotypic differences in molecular clock genes.

### Reporting summary

Further information on research design is available in the Nature Portfolio Reporting Summary linked to this article.

### Data availability

All data are contained within the manuscript and Supplementary files. The source data behind Figs. 2–7 can be found in the Supplementary Data file. All other data are available from the corresponding author on reasonable request.

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

## Acknowledgements
This work was funded by grants R01 AR076924 (N.Y.K.) and by Texas Woman's University Research Enhancement Program Award #3566 (H.B.E.). We are truly grateful to Dr. Melissa Harris, Associate Professor in the Department of Biology at the University of Alabama at Birmingham College of Arts and Sciences for teaching us about different types of hair and how to analyze them.

## Author contributions
Kelli R. Goggans: data curation, formal analysis, investigation, methodology, validation, visualization, writing—original draft, writing—review and editing. Olga V. Belyaeva: formal analysis, investigation, methodology, validation, visualization, writing—original draft. Alla V. Klyuyeva: investigation, methodology, validation, visualization. Jacob Studdard: investigation, methodology, validation, visualization. Aja Slay: investigation, methodology, validation, visualization. Regina B. Newman: data curation, formal analysis, investigation, methodology, validation, visualization, writing—original draft, writing—review and editing. Christine A. VanBuren: data curation, formal analysis, investigation, methodology, validation, visualization, writing—original draft, writing—review and editing. Helen B. Everts: conceptualization, data curation, formal analysis, project administration, supervision, validation, visualization, writing—original draft, writing—review and editing. Natalia Y. Kedishvili: conceptualization, data curation, formal analysis, project administration, supervision, validation, visualization, writing—original draft, writing—review and editing.

## Competing interests
The authors declare no competing interests.
