## [Peer Review File · Communications Biology]

Reviewers' comments:

Reviewer #1 (Remarks to the Author):

The manuscript from Goggans et al. reports novel findings regarding retinoid metabolism and actions in mouse skin and how this affects hair follicle (HF) development and hair composition. The authors studied double knockout mice that they generated, ones lacking expression of both epidermal retinol dehydrogenase 2 (RDHE2) and RDHE2-similar (RDHE2S), for all of their investigations. The authors have provided a considerable amount of new data that is needed for understanding HF development and more broadly skin biology. The authors data convincingly establish that RDHE2 and RDHE2S catalyze the rate limiting step for all-trans-retinoic acid (ATRA) synthesis in the HF resulting in accelerated anagen, slower catagen, altered HF composition, reduced levels of HF stem cell markers and dysregulated circadian clock gene expression in a sex-dependent manner. The author's data provide definitive and clear understanding of ATRA and RDHE2/RDHE2 actions in HF biology. Given the considerable disagreement in the skin field regarding these issues, disagreement that has existed for the last 20 years, this manuscript must be viewed as being a very significant one that considerably benefits understanding of skin biology.

The manuscript is well written and interesting. It was a pleasure to read this informative article. The studies are rigorous and the data are compelling. The authors have been conservative in interpreting their data. The findings will be of interest to a broad readership including investigators interested in skin biology, hair growth, cell proliferation and differentiation, circadian rhythms, short chain dehydrogenase/reductase actions and retinoid biology and actions.

Overall, this is a strong and significant manuscript, however there are a few points the authors should consider. These include:

Figure 1 shows retinyl esters being loaded into lipid droplets by LRAT. However, no indication is provided that retinol can leave the lipid droplets too. Although the specific identity of the enzyme responsible for catalyzing retinyl ester hydrolysis is unknown, it does happen. And, this may be important for understanding retinoid actions in the HF. The authors should add something to Figure 1 and/or the text surrounding the figure regarding ester hydrolysis.

On page 9, the authors need to provide a reference for "...Peterson's modification..."

Figure 2, panel f, provides data for only 3 postnatal days. All of the other panels provide data for 6 postnatal days. Why was this? This should be noted in the legend and/or text. Although this is unlikely to change the interpretation of these data, this needs to be added.

On each of the panels composing Supplemental Figure 2, a number appears immediately to the left of each image. What does this indicate? Presumably mouse number? Possibly this should be removed or defined.

Reviewer #2 (Remarks to the Author):

In this study by Goggans et al, the authors argued that ATRA signaling and its metabolizing genes change during HF cycle in a sex-dependent manner. To find functional relationship, they induced the deletion of RDHEs, which is a rate-limiting step of ATRA biosynthesis, and showed a couple of HF phenotypes including accelerated anagen, slower catagen, altered HF composition, reduced levels of HF stem cell markers. However, the overall results are not solid and convincing. Particularly, the circadian parts are suffered from the lack of convincing data. Overall, even though the study would provide interesting insights of retinol dehydrogenases functions, the reviewer has some concerns about misleading.

There are several specific comments.

- 1) In Figure 2a, the authors argue that the loss of RDHE induces variation of hair stages. However, there is no actual data, but only schematic diagram. To convince readers, the authors should provide proper real data with statistical tests.
- 2) The author's claims of sex-dependent manner were inconsistent. The authors noted more hair cycles were affected in male mouse in Figure 2a, whereas in subsequent data, the authors mentioned fluctuations in RDHE and target genes only in female mouse. Also, there did not appear to be a correlation between RDHE and HF cycles, as only Figure 2c showed a pattern and no fluctuations were found in other data.
- 3) The conclusion of this study was too over-stated without proper supporting data. The authors should have analyzed more PD time points (including first HF cycle and after PD50) in addition to the second synchronized HF cycle. Moreover, to show the correlation between RDHE and target genes, the study largely depends on qPCR analysis of bulk skin tissues. To get more spatial information, the authors should perform more IF, IHC, or ISH analysis in RDHE KO conditions. Finally, to pinpoint the tissue specific function of RDHE gene, tissue specific conditional KO mouse study would be required.
- 3) In Figure 2g, 2h, 2i and 2j, the expression patterns of RDHE mRNA and protein showed discrepancy. To reconcile this, the authors may perform additional assays that test post-transcriptional or post-translational changes in RDHE mRNA and protein.

Minor issues.

- 1) The authors showed a schematic model for the retinoid metabolism and signaling in Figure 1. However, the picture needs to be reconstructed to make it easier to understand.
- 2) It would be better to display the follicle images at PD40 rather than at PD35 in Figure 2b, as the hair cycle differences are more distinct at PD40 than at PD35 in male mice.
- 3) In Figure 2g and 2h, the normalized protein control should be matched.
- 4) In Figure 4C, the quality of IHC images for stem cell markers were poor. The images should be

replaced.

5) In Figure 5a and 5b, the authors showed images of the mouse overall fur. The authors also need to present images of each hair follicle. And, the authors mentioned the increase in only awl hair type in DKO mouse and no change in other hair types, so can the authors assess the total number of hair follicles in the DKO mouse compared to WT mouse?

6) In Figure legends of Figure 6a and 6b, the authors described these data was generated using same samples as Figure 2 (n=3 per PD). However, in the images, they were not presented by PD time points. This should be checked. Moreover, as mentioned above, the authors should also show data at various PD time points in addition to the second HF cycle.

Reviewer #3 (Remarks to the Author):

The study is a comprehensive and sex-specific analysis of ATRA regulation and signaling across all stages of the hair follicle cycle. The authors found that ATRA signaling occurs in a sex specific cyclic pattern and that the retinol dehydrogenase epidermal proteins (RDHEs) are essential for maintaining the cycle expression of ATRA-regulated transcripts. This is a novel, well designed and robust study that has identified a key role for the RHDEs in skin retinoid homeostasis, hair type composition and sex-specific hair follicle cycle progression. The data provided is convincing and supports the conclusions discussed by the authors. The statistical analyses performed are appropriate for the experimental design and the methodology is described in appropriate detail.

We would like to thank the reviewers for their positive comments and constructive critique. We have revised the manuscript to address the reviewers' comments. The detailed responses are listed below.

Reviewer 1

1. Figure 1 shows retinyl esters being loaded into lipid droplets by LRAT. However, no indication is provided that retinol can leave the lipid droplets too. Although the specific identity of the enzyme responsible for catalyzing retinyl ester hydrolysis is unknown, it does happen. And, this may be important for understanding retinoid actions in the HF. The authors should add something to Figure 1 and/or the text surrounding the figure regarding ester hydrolysis.

Thank you, this process has been added to Figure 1 to enhance clarity.

2. On page 9, the authors need to provide a reference for "...Peterson's modification..."

This reference was added.

3. Figure 2, panel f, provides data for only 3 postnatal days. All of the other panels provide data for 6 postnatal days. Why was this? This should be noted in the legend and/or text. Although this is unlikely to change the interpretation of these data, this needs to be added.

We agree. This data has been added to ensure our results are thorough.

4. On each of the panels composing Supplemental Figure 2, a number appears immediately to the left of each image. What does this indicate? Presumably mouse number? Possibly this should be removed or defined.

This number acts as an identification number internally but has been removed from the publication as recommended.

Reviewer 2

1. In Figure 2a, the authors argue that the loss of RDHE induces variation of hair stages. However, there is no actual data, but only schematic diagram. To convince readers, the authors should provide proper real data with statistical tests.

We expanded Figure 2a and b into Figure 2. We ran statistical analysis and show images of hair follicles to highlight the statistical differences.

2. The author's claims of sex-dependent manner were inconsistent. The authors noted more hair cycles were affected in male mouse in Figure 2a, whereas in subsequent data, the authors mentioned fluctuations in RDHE and target genes only in female mouse. Also, there did not appear to be a correlation between RDHE and HF cycles, as only Figure 2c showed a pattern and no fluctuations were found in other data.

After running statistical analysis on hair follicle stage, we see that the hair cycle is accelerated in both female and male mice but at different stages of postnatal development. In females, we see genotype differences in hair cycle stage at PD26 and PD30, when RDH activity and protein levels are the highest. In males, the genotype differences were seen at PD48 and later, when protein levels of RDHE2 peaked. The activity and RDHE2 and RDHE2S expression levels were performed in isolated microsomes, which included multiple cell types that may be masking differences.

3. The conclusion of this study was too over-stated without proper supporting data. The authors should have analyzed more PD time points (including first HF cycle and after PD50) in addition to the second synchronized HF cycle. Moreover, to show the correlation between RDHE and target genes, the study largely depends on qPCR analysis of bulk skin tissues. To get more spatial information, the authors should perform more IF, IHC, or ISH analysis in RDHE KO conditions. Finally, to pinpoint the tissue specific function of RDHE gene, tissue specific conditional KO mouse study would be required.

Thank you, we have extended histological analysis of the hair cycle past PD50 in males to adequately capture the telogen to anagen transition. This study already encompasses not only characterization of RDHEs at the level of mRNA, protein, and enzymatic activity in wildtype mice across all stages of the hair cycle, but also in an RDHE2/RDHE2S double knockout mouse model. Additionally, we have placed emphasis on analyzing both male and female mice and providing additional analysis interrogating the retinoic acid biosynthesis and degradation pathways, assessing changes in hair follicle stem cell markers and molecular clock genes, and characterization of hair type composition. To begin the hair cycle analysis earlier would not allow for sufficient analysis such as performed, as mice prior to PD26 are simply too small for adequate collection of skin. These studies aimed to minimize mouse numbers by fully utilizing dorsal skin from a single mouse for various studies, and this simply would not be possible in younger, smaller, mice. Additionally, the aims of this study focused on whole-skin analysis. We explain the limitations of this approach in the publication, but without a general understanding of the role of RDHEs across the hair cycle, further spatial interrogation could have been extensive without yielding adequate results for the cost of mice, materials, and time. Previous published research already indicated RDHEs are localized to the hair follicle and sebaceous gland, indicating that results gathered from whole skin would be indicative of the role of RDHEs within these areas. Additionally, further research is planned to further interrogate RDHEs in a spatiotemporal manner, but do not fit within the scope of this paper. Finally, a tissue-specific conditional KO mouse is not readily available and beyond the scope of this study. Single knockouts of RDHE2 and RDHE2s (RDHEs reported in the publication) were found to lack phenotypes seen in the *double* knockout mouse model. Our DKO mice were created via CRISPR-Cas9 due to the close proximity of *Rdhe2* and *Rdhe2s*, which is not compatible with tissue-specific KOs.

4. In Figure 2g, 2h, 2i and 2j, the expression patterns of RDHE mRNA and protein showed discrepancy. To reconcile this, the authors may perform additional assays that test post-transcriptional or post-translational changes in RDHE mRNA and protein.

The study already reports novel information for RDHEs at the level of mRNA, protein, and enzyme activity. Doing additional post-transcriptional analyses is beyond the scope of this study.

Minor:

5. The authors showed a schematic model for the retinoid metabolism and signaling in Figure 1. However, the picture needs to be reconstructed to make it easier to understand.

We revised this figure.

6. It would be better to display the follicle images at PD40 rather than at PD35 in Figure 2b, as the hair cycle differences are more distinct at PD40 than at PD35 in male mice.

We agree and have replaced the figures accordingly. They are in the new Figure 2.

7. In Figure 2g and 2h, the normalized protein control should be matched.

We are unsure what reviewer 2 means by 'matched'. We have provided adequate explanation to the unfortunate use of two normalization techniques due to the removal of our primary microsomal marker, p450 reductase antibodies, from the market.

8. In Figure 4C, the quality of IHC images for stem cell markers were poor. The images should be replaced.

We have made efforts to replace them with even better images where possible.

9. In Figure 5a and 5b, the authors showed images of the mouse overall fur. The authors also need to present images of each hair follicle. And, the authors mentioned the increase in only awl hair type in DKO mouse and no change in other hair types, so can the authors assess the total number of hair follicles in the DKO mouse compared to WT mouse?

Assessment of the hair follicles in RDHE double KO mice was performed in a previous study and showed no significant variations from wildtype hair follicles besides enlargement of the associated sebaceous gland. Additionally, skin sections with numerous hair follicles are presented in Figure 2 and Supplementary Figure 1 across all assessed hair cycle time points, genotypes, and sexes.

10. In Figure legends of Figure 6a and 6b, the authors described these data was generated using same samples as Figure 2 (n=3 per PD). However, in the images, they were not presented by PD time points. This should be checked. Moreover, as mentioned above, the authors should also show data at various PD time points in addition to the second HF cycle.

We have removed this comment from the legends of Figures 6a and 6b for clarity. PD50 was chosen to demonstrate potential differences in molecular clock genes as previous literature (Lin 2017) demonstrated that the molecular clock fluctuates in amplitude and is strongest at PD50. Additionally, the same mouse samples were used for the sake of conservation of mice. Additional characterization of molecular clock genes across the hair cycle from a single 24-hour time point would be disingenuous in

the assessment of the molecular clock, which would require at least 6 samples from the 24-hour cycle. The data presented here is merely our additional efforts to identify potential mechanisms involved with RDHEs and provide a springboard for further research.

Reviewers' comments:

Reviewer #1 (Remarks to the Author):

The authors have addressed well my earlier concerns. I have no new or additional concerns.

Reviewer #2 (Remarks to the Author):

The revised manuscript has been improved. However, this reviewer thinks there are a couple of issues that should be addressed before publication. Below is the suggestions and concerns.

1. In Figure 2, the authors revised figures and now clearly showed that male dKO mice exhibit accelerated hair cycle entry during 2nd telogen (around P50). But because it is hard to the identity of each panel (Fig 2c - 2v), it would be good if each panel has short label. For example, Fig 2c can be labeled "WT, P25" (not sure if it is even true according to legendd). Now it is very hard to see which one is which.

2. Reading the rebuttal and methods section again, the reviewer has just realized that the authors took a small piece of skin from a same mouse in different timepoints and analyzed them in a kind of longitudinal manner. However, the skin prep biopsy is an injury and triggers wound healing processes which significantly affect hair cycle regulation in mice. Sorry for this late issue raising, but the study should be re-interpreted in a view of skin wound healing processes.

We addressed comments from Reviewer 2 as follows.

1. In Figure 2, the authors revised figures and now clearly showed that male dKO mice exhibit accelerated hair cycle entry during 2nd telogen (around P50). But because it is hard to the identity of each panel (Fig 2c - 2v), it would be good if each panel has short label. For example, Fig 2c can be labeled "WT, P25" (not sure if it is even true according to legend). Now it is very hard to see which one is which.

Thank you for your suggestion. We have labeled the panels accordingly.

2. Reading the rebuttal and methods section again, the reviewer has just realized that the authors took a small piece of skin from a same mouse in different timepoints and analyzed them in a kind of longitudinal manner. However, the skin prep biopsy is an injury and triggers wound healing processes which significantly affect hair cycle regulation in mice. Sorry for this late issue raising, but the study should be re-interpreted in a view of skin wound healing processes.

Sorry for not being explicitly clear on this point. We have further clarified in the Methods section that the skin taken from an individual mouse was used for multiple assays at one timepoint and not multiple timepoints. The wound healing processes were not triggered because all skin was taken right after sacrifice.